# Unlocking the Potential of Camel Milk-Derived Exosomes as Novel Delivery Systems: Enhanced Bioavailability of ARV-825 PROTAC for Cancer Therapy

**DOI:** 10.3390/pharmaceutics16081070

**Published:** 2024-08-15

**Authors:** Aakash Nathani, Mounika Aare, Li Sun, Arvind Bagde, Yan Li, Arun Rishi, Mandip Singh

**Affiliations:** 1College of Pharmacy and Pharmaceutical Sciences, Florida A&M University, Tallahassee, FL 32307, USA; aakash1.nathani@famu.edu (A.N.); mounika1.aare@famu.edu (M.A.); arvind.bagde@famu.edu (A.B.); 2Department of Chemical and Biomedical Engineering, FAMU-FSU College of Engineering, Florida State University, Tallahassee, FL 32310, USA; li.sun@med.fsu.edu (L.S.); yli@eng.famu.fsu.edu (Y.L.); 3Department of Biomedical Sciences, College of Medicine, Florida State University, Tallahassee, FL 32304, USA; 4Department of Oncology, John D. Dingell VA Medical Center, Wayne State University School of Medicine, Detroit, MI 48201, USA; rishia@karmanos.org

**Keywords:** oral bioavailability, ARV-825, camel milk exosomes, anticancer

## Abstract

This study investigates the use of camel milk-derived exosomes (CMEs) as carriers for ARV-825, an anticancer agent targeting bromodomain-containing protein 4 (BRD4), in oral chemotherapy. CMEs were isolated and characterized, and ARV-825-loaded CME formulations were prepared and evaluated through various in vitro and in vivo tests. The ARV-825-CME formulation exhibited an entrapment efficiency of 42.75 ± 5.05%, a particle size of 136.8 ± 1.94 nm, and a zeta potential of −32.75 ± 0.70 mV, ensuring stability and sustained drug release. In vitro studies showed a 5.4-fold enhancement in drug release kinetics compared to the free ARV-825 solution. Permeability studies indicated a 3.2-fold increase in apparent permeability, suggesting improved cellular uptake. Cytotoxicity assays demonstrated potent anticancer activity, with IC50 values decreasing by 1.5 to 2-fold in cancer cell lines SF8628 DIPG and H1975R (resistant to Osimertinib). In vivo pharmacokinetic studies in Sprague-Dawley rats revealed superior systemic absorption and bioavailability of ARV-825 from CMEs, with a 2.55-fold increase in plasma concentration and a 5.56-fold increase in AUC. Distribution studies confirmed absorption through the ileum. This research highlights the potential of CMEs as a promising delivery platform for ARV-825, enhancing its therapeutic efficacy and offering a novel approach to cancer treatment.

## 1. Introduction

Cancer remains a leading global health challenge, second only to heart disease, with one in six individuals succumbing to it, according to the WHO [1,2]. Despite significant investment in anticancer drug research, the success rate from discovery to FDA approval is low, with only 10–12% of drugs in Phase I trials receiving approval [3,4,5,6]. A major obstacle is the poor water solubility of many anticancer compounds, which hampers their pharmacokinetic and pharmacodynamic profiles. Oral anticancer drugs, despite their convenience, often suffer from low solubility, impacting their efficacy and increasing toxicity risks [7,8,9,10]. Nevertheless, amidst these challenges, the dominance of oral anticancer drugs in the pharmaceutical landscape for over two decades underscores a notable shift in therapeutic approaches [11]. However, several challenges limit the achievement of desirable pharmacokinetics of anticancer drugs, which in turn limits the pharmacodynamics of orally administered, arise from drug physicochemical properties such as poor water-solubility [12]. Oral chemotherapy has the potential to significantly improve patients’ conditions while allowing for therapeutic dose regulation with minimal to moderate impact on off-target tissues [13]. The only off-target effects of oral delivery majorly impact the gastrointestinal system due to direct contact with the drug and hepatic portal system, which can cause toxicities in the stomach, intestine, and liver. For example, methotrexate is widely used in oral chemotherapy and is known for its hepatotoxic reactions [14]. This has posed a substantial obstacle to the translation of powerful therapeutic candidates, which have great potential by nature but lack the ability to show a discernible clinical impact because of dose-related toxicities or dose-limited efficacies brought on by off-target effects [15]. However, off-targeting can be drastically reduced with the administration of drugs orally using drug delivery systems that can minimize the accumulation of drugs in organs and tissues other than the target site. This trend suggests that earlier apprehensions regarding the feasibility and efficacy of oral anticancer therapy may have been exaggerated. The proliferation of approved oral anticancer drugs signals a promising trajectory in cancer treatment, albeit with persistent hurdles to overcome. In conclusion, while the landscape of anticancer drug development has witnessed significant advancements, formidable challenges persist, necessitating continued innovation and concerted efforts [16]. Addressing issues such as poor solubility through targeted research and technological innovation holds the promise of unlocking new avenues for improving cancer treatment outcomes and reducing the global burden of this devastating disease. Hence, there is a need for novel delivery systems, such as exosomes, to enhance oral bioavailability and safety of anticancer agents.

Exosomes are cellular-derived membrane vesicles that are found in almost all physiological fluids, such as blood, urine, saliva, and milk [17,18,19], and contain a variety of biologically significant molecules, including proteins, lipids, DNA, mRNA, microRNA, and others [20]. Their high biocompatibility and biodegradability make them an appealing alternative tool for therapeutically relevant drug delivery [20,21]. Many investigators have used exosomes that are derived from cell cultures, but the yield from this source is low and needs bioreactors and a lot of media to be processed, which may cause FDA regulatory issues [22,23]. Milk is a unique, natural, and inexpensive source of exosomes that can be easily processed in liters of volume in a short time. Moreover, milk exosomes exhibit remarkable resilience to the harsh environment of the gastrointestinal tract. In vitro studies have showcased their strong attraction to epithelial cells, their ability to be taken up by cells through endocytosis, and, significantly, their promise for oral administration as a potential delivery mechanism [24]. Milk exosomes can transport hydrophilic and lipophilic drugs because their membrane contains both hydrophilic and hydrophobic components [25]. Milk exosomes are increasingly recognized as a promising alternative to exosomes sourced from cell cultures. Their emergence as a viable option stems from their notable potential for oral drug delivery, which underscores their importance in the field [26]. Unlike protein components of cancer cells and exosomes derived from other sources, which can cause immune responses when administered systemically, milk exosomes do not [27]. Milk exosomes have been isolated from humans, camels, rats, horses, sheep, pandas, porcine, cattle, yaks, and goats [28]. However, camel milk-derived exosomes (CMEs) have been reported to possess anticancer activity by increasing apoptosis [29,30]. For this added advantage, we chose camel milk as the source of exosomes for loading our therapeutic agent ARV-825. From this point, the term exosome in methods, results, and discussion refers to CMEs and will be used interchangeably.

Recent discoveries in cancer research have pinpointed epigenetic proteins as promising targets for anticancer therapies. One notable group within this category is the bromodomain and extra terminal (BET) protein family, which has garnered significant interest. This interest stems from its pivotal function in recruiting enzymes that regulate chromatin, as well as its ability to interpret chromatin structure, thereby influencing gene expression crucial in cancer progression, such as in gliomas and carcinomas [31,32]. Bromodomain-containing protein 4 (BRD4) is a member of the BET family of proteins, consisting of two bromodomains (BD) and one extra terminal (ET) [33,34]. BRD4 is highly expressed in most human cancers and can be an oncotarget [35,36]. ARV-825 is a new inhibitor developed using PROTAC technology that combines OTX015 with the E3 ligase cereblon (CRBN). ARV-825 administration causes BRD4 recruitment to cereblon, which results in BRD4 degradation that is rapid, efficient, and prolonged [37]. By directly recruiting BRD4 to the E3 ubiquitin ligase cereblon, it facilitates the swift, effective, and enduring degradation of BRD4 protein. Research indicates that ARV-825 exhibits superior efficacy compared to conventional small molecule BRD4 inhibitors in dampening BRD4 signaling. This leads to robust and sustained suppression of cancer cell activity and triggers significant apoptosis, highlighting its potential as a highly impactful therapeutic agent [34,37,38]. However, ARV-825 is poorly water-soluble [39]. Therefore, the objectives of this study were to develop ARV-825 loaded CME (ARV-825-CME) formulation and evaluate it in various in vitro and in vivo studies. This study will demonstrate for the first time the potential of CMEs as oral drug delivery vehicles for chemotherapeutic drugs with enhanced bioavailability. To investigate this potential, CMEs were isolated and characterized, ARV-825-loaded CME formulations were prepared and optimized, and their stability, drug release kinetics, cellular uptake, cytotoxicity, in vivo pharmacokinetics, and tissue distribution were systematically studied.

## 2. Materials and Methods

### 2.1. Materials

#### 2.1.1. Materials

ARV-825 was purchased from MedChem Express (Monmouth Junction, NJ, USA). Camel (*Camelus dromedaries*) milk was obtained from Desert Farms (Irvine, CA, USA). Ethanol (Sigma Aldrich, St. Louis, MO, USA), bovine serum albumin (Genesee Scientific, Morrisville, NC, USA), probe sonicator (Sino Sonics, Wuxi, China), kolliphor EL (BASF Pharma, Florham Park, NJ, USA), sucrose (Sigma Aldrich, St. Louis, MO, USA), were obtained from various sources as described. The dialysis membrane tubing was purchased from Sigma-Aldrich (St. Louis, MO, USA). The SYTO RNASelect green fluorescence stain was obtained from ThermoFisher Scientific (Hanover Park, IL, USA). The H1975 cells (with EGFR mutations L858R) were purchased from ATCC and were made resistant to Osimertinib (H1975R) in the laboratory (Gaithersburg, MD, USA).

#### 2.1.2. Animals

Sprague Dawley (SD) rats, procured from Charles River Laboratories in Wilmington, Massachusetts, were utilized for conducting in vivo pharmacokinetic (PK) studies. These rats were housed in cages furnished with appropriate bedding, ensuring controlled environmental parameters, including a temperature range of 22 ± 2 °C, a consistent 12:12 light-dark cycle, and a relative humidity maintained at 50 ± 15%. The housing facilities were situated within Florida A&M University’s animal facilities, where strict adherence to the guidelines outlined in the “Guide for the Care and Use of Laboratory Animals” and the criteria set forth by the Association for Assessment and Accreditation of Laboratory Animal Care (AAALAC) was ensured. Before the commencement of any experimental procedures, the rats underwent a one-week acclimatization period within the laboratory settings. Furthermore, the Institutional Animal Care and Use Committee (IACUC) of Florida Agricultural and Mechanical University granted approval for the animal protocol (022-08) implemented in this study.

### 2.2. Methods

#### 2.2.1. Exosome Isolation and Purification

Exosomes were isolated using a method of differential ultracentrifugation, previously described in detail [30,40]. In summary, milk samples underwent centrifugation twice: initially at 3000× *g* at 4 °C for 30 min to remove fat and debris, followed by centrifugation at 10,000× *g* at 4 °C for 30 min to obtain clear milk serum. Subsequently, exosome pellets were isolated from the milk serum by subjecting them to centrifugation at 100,000× *g* at 4 °C, repeated twice for 60 min each time, utilizing the Optima L-90K ultracentrifuge from Beckman Coulter, Brea, CA, USA. The resulting exosome pellets were collected and dissolved in PBS to ensure a homogeneous suspension and subsequently stored at −80 °C in aliquots until required for further use.

#### 2.2.2. Preparation and Optimization of ARV-825 Loaded Exosomes

To prepare ARV-825 loaded exosomes for administration, ARV-825 was first dissolved in a solution consisting of 5% *v*/*v* ethanol and 10% *w*/*v* PEG 400. PBS buffer containing CMEs (with a particle number of 7.1 × 10^11^/mL, equivalent to a 2-fold ARV-825 concentration based on protein weight) was then added to this solvent mixture and gently mixed. Emulsification was facilitated by the addition of Kolliphor EL (at a concentration of 10%), followed by vertexing for 1 min. The formulation underwent further processing by being sonicated in a bath sonicator, followed by probe sonication (three cycles at an amplitude of 20%) lasting 30 s each, with a 2-min rest period in between cycles. Subsequently, the formulation was left to shake overnight at 4 °C. To ensure the removal of any unentrapped drugs, the final formulation underwent ultracentrifugation at 100,000× *g* for 120 min. The resulting supernatant was discarded, while the pellet containing the encapsulated ARV-825 was retained. PBS was then added to dissolve the pellet, which was left to shake overnight at 4 °C. The final formulation was subjected to analysis using the LC-MS method to quantify the extent of drug entrapment, thereby confirming its readiness for use in subsequent studies.

#### 2.2.3. Control ARV-825 Solution Preparation

Briefly, ARV-825 was solubilized in ethanol and PEG 400 mixture as described above. The solution was then sonicated using a bath sonicator, filtered, and analyzed using an established LC-MS method to determine the concentration of ARV-825.

#### 2.2.4. Exosome Characterization

##### Entrapment Efficiency and Nanoparticle Tracking Analysis (NTA)

In order to extract the ARV-825, the entrapment efficiency was measured by diluting the ARV-825-CME with methanol (1:1). Following a 5-min vortex, a 5-min bath sonication, and a 15-min centrifugation at 14,000 rpm at 4 °C, the sample was analyzed using the LC-MS method. The percentage entrapment efficiency (% EE) was calculated by using the following equation:$$\%\;\mathrm{EE}\;=\frac{\mathrm{Amount}\;\mathrm{of}\;\mathrm{drug}\;\mathrm{entrapped}}{\mathrm{Initial}\;\mathrm{amount}\;\mathrm{of}\;\mathrm{drug}\;\mathrm{taken}}\times 100$$

Using NTA (ZetaView^®^ TWIN PMX-220, Ammersee, Germany), as previously used in scatter and fluorescent modes, the average particle size (z-average), concentration, and zeta potential of the CMEs and ARV-825-CME formulations were examined using the dynamic light scattering (DLS) approach at 25 °C and a 90° scattering angle to characterize total particles and purity [41]. The samples were then diluted with particle-free water at a 1:1000 ratio and measured in triplicate. To quantify fluorescence, exosomes were stained with CD9/CD63/CD81 antibodies that had been fluorescently labeled in accordance with the protocol developed by Particle Metrix (Ammersee, Germany) using Alexa 488 fluorochrome in fluorescent mode. This allowed for the characterization of the exosome membrane vesicles, and ZetaView Analysis software (version 8.05.12 SP1) was used to process the data [42]. The following formula was used to determine the purity percentage:$$\%\;purity=\frac{Conc.\;in\;fluorescent\;mode}{Conc.\;in\;scatter\;mode}$$

##### Western Blotting for Exosomal Markers in CMEs

The radio-immunoprecipitation assay (RIPA) buffer, which included 50 mM Tris pH 8.0, 0.1% SDS, 1.0% Triton X-100, 150 mM sodium chloride, 0.5% sodium deoxycholate, 0.1% SDS, and 1× Thermo Scientific Halt Protease and Phosphatase Inhibitor Cocktail, was used to lyse CMEs. The lysing process occurred on ice for 20 min, followed by centrifugation at 14,000 rpm for 20 min at 4 °C. Upon collecting the supernatant, the protein concentration was determined using the bicinchoninic acid assay (Smith test). Subsequently, the protein lysate was denatured in 3× Laemmli sample buffer at 90 °C for 7 min. Proteins were electrophoretically transferred using the Trans-Blot^®^ TurboTM Transfer System from Bio-Rad, Hercules, CA, USA, from an SDS-PAGE gel to a nitrocellulose membrane. After transfer, the membrane was blocked with PBS containing 0.1% Tween 20 and 3% *w*/*v* BSA for an hour at room temperature. After applying primary antibodies to the membrane and letting them sit overnight at 4 °C, the antibodies were diluted to a concentration of 1:1000 in the blocking solution. After being incubated at room temperature for two hours with secondary antibodies conjugated with horseradish peroxidase, the blots were subjected to three 5-min-long washes in PBST (PBS containing 0.1% Tween-20). The Super Signal West Pico Chemiluminescent substrate was then added to the blots, and pictures were taken with the Bio-Rad Chemidoc Instrument. To quantify the data, densitometry scanning of the immunoblots was carried out using NIH ImageJ software (version 1.54) [30,41].

##### Transmission Electron Microscopy (TEM) of Exosomes

TEM analysis was performed to examine the exosome morphology of ARV-825-CME. Briefly, exosomes were thawed from −80 °C and then diluted 1:10 using PBS. A copper grid was coated with diluted exosomes and stained using the negative stain, 1% *v*/*v* uranyl acetate, in double-distilled water. The Hitachi HT 7800 TEM was used to observe the stained grids [43]. 

##### Proteomic Analysis of CME Protein Cargo

Exosomes from different types of milk—camel, cow, and goat—were isolated, and their protein content was analyzed. To begin, the exosomal proteins were extracted and quantified. Using S-trap micro columns, up to 20 µg of proteins were isolated, ensuring a representative sample for subsequent analysis. These proteins underwent a series of steps, including alkylation and digestion on the column, following standardized protocols provided by the manufacturer. The processed protein samples, maintained in triplicate for each milk group, were then prepared for analysis at Florida State University’s Translational Science Laboratory.

Employing high-resolution mass spectrometry on the Thermo Q Exactive HF instrument, the protein content of the exosomes was characterized. The resulting raw data files were subjected to rigorous analysis using Proteome Discoverer 2.4, which employed multiple search engines—SequestHT, Mascot, and Amanda—to ensure comprehensive identification of proteins. Validation of both peptide and protein identities was performed using Scaffold (version 5.0), a powerful tool for proteomic data validation. For peptide identity validation, the Scaffold local false discovery rate (FDR) algorithm was utilized, setting a stringent threshold of greater than 99.0% probability for acceptance. Similarly, protein identity validation required a probability level exceeding 99.0% and a minimum of two recognized peptides.

Quantification data obtained from the analysis was processed using Express Analyst with default settings, allowing for the identification of differentially expressed proteins (DEPs) using DEseq2. Furthermore, heat maps and volcano plots were generated using the same online tool to visually represent the global differences in gene expression, providing insightful visualizations of the findings [44]. This comprehensive approach enabled a thorough examination of the protein content of exosomes derived from camel, cow, and goat milk, shedding light on potential differences and contributing to our understanding of milk-derived exosomal biology.

#### 2.2.5. In Vitro Release Study

An in vitro drug release study was carried out using a modified dialysis bag method, as described in an earlier report [45]. Briefly, ARV-825-CME (1 mL equivalent to 1.25 mg of ARV-825) was sealed in a preactivated dialysis membrane bag with a molecular weight cut-off (MWCO) of 12 kDa (Sigma-Aldrich, St. Louis, MO, USA) and immersed in 100 mL of release media (PBS, pH 7.4, 0.5% tween 80, 5% ethanol) at 37 ºC with a magnetic stirrer at 75 rpm. At predetermined intervals, i.e., 0.5, 1, 2, 4, 8, 12, 24, 48, and 72 h, 1 mL samples were removed from the receiver compartment, replaced with fresh medium to maintain sink conditions, and analyzed for ARV-825 using an already established LC-MS method. A release study was also conducted in a pH 1.2 acidic buffer. Samples were collected at 0.5, 1, 1.5, and 2 h and analyzed using LCMS.

#### 2.2.6. In Vitro Permeability Studies

MDCK cells were grown in an EMEM medium with 10% FBS and 1% PSN added. After the cells reached 80% confluence, trypsinization and a PBS wash were performed on them. In a 24-well plate, cells were seeded at a density of 33,000 cells/well in 100 μL cell solution using a polyester membrane-covered Costar^®^ Trans well^®^ permeable support with a 3.0 μm pore size and 6.5 mm insert diameter. The plates were kept at 37 °C with 5% CO_2_ in an incubator. A Millicell ERS-2 Epithelial volt-ohm meter (Millipore, Billerica, MA, USA) was used to measure transepithelial electrical resistance (TEER) in order to evaluate tight junction development and monolayer integrity. The apical compartments were washed with HBSS once the Trans-well plate was completely confluent. The media of the basolateral compartment was replaced with HBSS. The plates were kept at 37 °C for 10 min for acclimatization. After that, ARV-825-CME and ARV-825 solution (150 µg/mL) were added to the apical compartment, and the plates were kept for agitation in a humidified chamber at 37 °C and 5% CO_2_. Samples were collected after 2 h and analyzed using the above-mentioned LC-MS method. The apparent permeability was calculated using the prescribed formula [46].
$$PAPP(cm/s)=\frac{dQ}{Ci\times dT\times A}$$
where dQ = amount of drug in receiving compartment (nmole), Ci = the initial concentration of drug (nmole/mL), T = time (s), and A = area of insert (cm^2^).

#### 2.2.7. Cytotoxicity Assay

##### 2D Viability Assay Using MTT

H1975R lung cancer cells with EGFR mutations (L858R) and SF8628 diffuse intrinsic pontine glioma (DIPG) cells with the histone H3.3 Lys 27-to-methionine (K27M) mutation were subjected to a concentration-dependent cytotoxicity assay. The cells were seeded in a 96-well plate at a density of 5000 cells per well and maintained at 37 °C with 5% CO_2_. The experimental media, which contained CMEs, ARV-825 solution, and ARV-825-CME, was added after 24 h and incubated for 48 h. After four hours of treating the cells with MTT (0.5 mg/mL) solution, the formazan crystals that were produced were dissolved in 100 µL of DMSO. A Tecan Infinite 200 PRO M Plex multimode microplate reader was used to measure the absorbance at 570 nm [47].

##### 3D Viability Assay Using MTT

3D spheroids were cultured in the laboratory using established protocols and a magnetic nanoshuttle system [41]. Briefly, SF8628 DIPG cells and H1975R cells were tagged with nanoshuttle solution (Greiner, Freickenhausen, Germany) at 10 μL for 10,000 cells and centrifuged at 800 rpm for 7 min three times. To allow the formation of 3D spheroids, 15,000 cells/well were seeded in a 96-well plate with a cell-repellent surface and kept on the magnetic drive. Over the course of five days, the plate was incubated at 37 °C and 5% CO_2_, with a change of media on the third day. The cells were treated with ARV-825 and ARV-825-CME formulation for 48 h, after which the media was replaced with a 0.5 mg/mL MTT solution and incubated for 4 h. The resulting formazan crystals were dissolved in 100 µL DMSO, and absorbance was measured at 570 nm as described for the 2D assay.

#### 2.2.8. Internalization of SYTO RNASelect Fluorescent Dye-Stained CMEs into H1975R Tumor

Briefly, PBS was used to dilute the SYTO RNASelect solution (5 mM) to make a stock solution with a concentration of 1 mM. To achieve a final concentration of 10 µM, 10 µL of stock solution was added for every milliliter of exosome suspension and well mixed by gentle pipetting. After 30 min of incubation at 37 °C, the mixture was subjected to ultracentrifugation in order to separate the unbound dye. For later usage, it was aliquoted as needed, stored at −80 °C, and shielded from light.

In vivo experiments were conducted to demonstrate the uptake of exosomes by tumor cells. Female BALB/c-nu/nu mice were acquired from Jackson Labs. H1975R cells (4 × 10^6^ cells) were combined with Matrigel (1:1) and subsequently injected subcutaneously into the right flank of each mouse. Solid tumors developed in the mice within 7 days, and treatment commenced once the tumor volume reached 100 mm^3^. Treatment initiation occurred 10 days after the implantation of tumor cells. The treatment regimens for mice bearing subcutaneous H1975R tumors resistant to Osimertinib comprised of (a) oral administration of PBS as a negative control, (b) exosomes alone, and (c) SYTO RNASelect dye-stained exosomes. There were 3 mice in each group. Tissue analysis was conducted on all mice at 4 h post-oral administration. Tumor cryosections (40 µm) were directly observed using a Nikon Eclipse Ti 100 inverted fluorescent microscope (Nikon Instruments, Inc., Melville, NY, USA), with the microscope configured to operate with a fluorescent filter at 490/530 nm (excitation/emission).

#### 2.2.9. In Vivo Pharmacokinetic (PK) Study

The animal experiments in this study were approved by the Institutional Animal Care and Use Committee (IACUC) at Florida A&M University in Tallahassee, FL, USA. All research followed the approved protocol (protocol approval # 022-08). ARV-825’s In vivo systemic absorption was assessed using male Sprague-Dawley rats weighing approximately 250 g. They were kept overnight fast before beginning the study. The animals were then divided into two groups: Group 1 received the ARV-825-CME formulation, and Group 2 received the control ARV-825 solution. Each group (5 animals per group) received a dose of 5 mg/kg via oral gavage. Blood samples were collected at time intervals of 0.5, 1, 2, 4, 6, 8, 12, 24, 48, and 72 h. To obtain plasma samples, whole blood samples were centrifuged at 4500 rpm at 4 °C. The resulting plasma samples were then kept at −80 °C until they were analyzed. All samples were then diluted 1:1 with acetonitrile and finally analyzed with the LC-MS method. Compartmental modeling of the pharmacokinetic data was performed using PK Solver.

#### 2.2.10. Distribution of ARV-825 in the Intestinal Sections and Plasma after Oral Administration of ARV-825-CME

The animal experiments in this study were also approved by the Institutional Animal Care and Use Committee (IACUC) at Florida A&M University in Tallahassee, FL, USA. All research followed the approved protocol (protocol approval # 022-08). ARV-825’s intestinal and plasma tissue distribution was assessed using the same animals and conditions as mentioned in the PK study. Animals received a single dose (5 mg/kg) of ARV-825-CME (stained) via oral gavage. Animals (3 at each time interval) were sacrificed at predetermined time intervals, and the duodenum, ileum, and jejunum were obtained from the intestinal tissues and washed with PBS solution using a syringe. The whole tissues were mixed with tissue protein extraction reagent, and exosome lysis buffer homogenized using a tissue blender and finally mixed with acetonitrile (1:1). After extraction, concentrations in different tissue sections were detected by LC-MS to study whether there were differences in amounts in distributions and to identify the part of intestine responsible for highest absorption. Plasma samples were also obtained and stored, as mentioned in the PK study, for further analysis.

#### 2.2.11. Stability Studies of ARV-825-CME at pH 1.2 and pH 6.8

ARV-825-CME was diluted (1:1) using phosphate buffers (pH 1.2 and pH 6.8) and were stored at 37 °C for 2 and 24 h, respectively. After that, samples were withdrawn to measure the zeta potential, particle size, and concentration of protein. The average particle size (z-average), concentration, and zeta potential of the ARV-825-CME were investigated using the dynamic light scattering (DLS) approach at 25 °C and a 90° scattering angle using NTA (ZetaView^®^ TWIN PMX-220) in scatter mode. All the samples were measured in triplicate using the ZetaView instrument after being diluted with particle-free water at a ratio of 1:1000.

For determining protein concentration, ARV-825-CME was lysed with RIPA buffer containing 150 mM sodium chloride, 1.0% Triton X-100, 0.5% sodium deoxycholate, 0.1% SDS, 50 mM Tris, pH 8.0, and 1× Thermo Scientific Halt Protease Inhibitor Cocktail. The samples were spun down at 14,000 rpm for 20 min after a 20-min ice lysis. After the supernatant was collected, the protein concentration was measured using the Smith assay (bicinchoninic acid assay).

#### 2.2.12. LC-MS Analysis for Quantification of ARV-825 in All the Studies Above

LC-MS analysis was performed utilizing equipment from Waters Technology Corporation, Milford, MA, USA, including a Waters e2695 separation module, a QDa detector, Aquity QDa, and a Waters 2998 photodiode array detector (PDA). The mobile phase comprised of 90:10 acetonitrile and water (with 0.1% formic acid), running at a flow rate of 1 mL/min with an injection volume of 30 μL. Column temperature was maintained at 40 °C. Elution of samples was conducted on a reverse phase C18 column (Nova-Pak^®^ 3.5 μm, 3.9 × 150 mm) with a guard column (Symmetry^®^, reversed phase, C18). The QDa settings included a mass range of 750–950 Da, positive scan mode, a cone voltage of 22 V, and capillary voltage settings of 1.5 kV for positive polarity and 0.8 kV for negative polarity. Selected ion recording (SIR) was performed at a mass of 923 Da in positive polarity mode with a cone voltage of 22 V.

#### 2.2.13. Statistical and PK Analyses

Raw data was presented as Mean ± standard deviation (SD) after at least three repetitions. The data was statistically analyzed using the student’s *t*-test for two-group comparisons with an unpaired experimental design and Welch’s correction or one-way ANOVA followed by “Bonferroni’s Multiple Comparison Test” for multiple variable comparisons. Significant differences between groups were defined as *p* < 0.05. Statistical analyses were carried out using GraphPad Prism 5.0 (GraphPad Software, Inc., San Diego, CA, USA). PKSolver was used to conduct noncompartmental analyses of in vivo pharmacokinetic data.

## 3. Results

### 3.1. Exosome and ARV-825-CME Characterization

CME NTA results in scatter mode showed a protein concentration of 6.2 ± 0.3 mg/mL (Bicinchoninic acid assay), average particle concentration of 8.70 × 10^11^/mL, particle size of 121.4 ± 2.17 nm, and zeta potential of −22.84 ± 0.54 mV. In fluorescent mode, the particle number was 7.92 × 10^11^/mL with a size of 117.6 ± 1.83 nm and a zeta potential of −32.54 ± 0.08 mV (Figure 1A–C). Furthermore, the percent purity of the particles was found to be 91.03 ± 1.76%. ARV-825-CME formulation was successfully formulated with a particle size of 136.8 ± 1.94 nm and a zeta potential of −32.75 ± 0.70 mV (Figure 1B,D). The formulation had a translucent physical appearance and no particle sedimentation. Western blotting was performed to determine known exosome markers following the separation of CMEs. It was confirmed that HSP90, Alix, CD81, and CD63 were expressed in the CMEs (Figure 1E). Entrapment efficiency was found to be 42.75 ± 5.05%. TEM images showed that the exosomes have intact vesicular membranes after freezing and thawing at −80 °C. It was also found that the particle sizes of CMEs and ARV-825-CME were between 100 and 130 nm. (Figure 1F,G).

Proteomics analysis of the CMEs (419 identified proteins) compared to the exosomes isolated from Cow (131 proteins) and Goat (133 proteins) milk was observed (Figure 2). The principal component analysis (PCA) plot showed the distinct cluster of the three groups, each of which had three replicates (Figure 2A). The heatmap indicated a similar composition for the Cow and Goat groups, but the CMEs had distinctly different protein expression levels (Figure 2B). The composition of the CME protein cargo is shown in Appendix A, containing 442 proteins. These proteins are related to cellular metabolism (e.g., G6PD, PFKL, etc.), EV biogenesis (e.g., TSG 101, RAB7A, CD63, etc.), cellular adhesion (e.g., CD44, ITGAM, ITGB2, etc.), and cytoskeleton (e.g., TUBB. ACTN4). The volcano plot of CMEs vs. Goat exosomes revealed 230 significantly upregulated proteins and 118 significantly downregulated proteins (Figure 2C). The volcano plot of CMEs vs. Cow exosomes revealed 359 significantly upregulated proteins and 113 significantly downregulated proteins (Figure 2D and Appendix A). Venn diagram showed the overlapped (38 proteins) and CME distinct proteins (356 proteins) (Figure 2E and Appendix A). For the overlapped proteins, the heatmap showed different expression levels for CMEs compared to the Goat and Cow groups, such as XDH, SLC34A2, MFGE8, PGLYRP1, KRT10, etc. (Figure 2F). These results suggest that exosomes isolated from different milk products have different protein cargo, and camel milk is the preferred source for this study to isolate exosomes. Control CMEs were compared to ARV-825-CME, and we identified 365 protein clusters, which is comparable to control CMEs, and 203 overlapped. In the top 20 protein list, there were 14 in common, and one particular protein, casein, was greatly reduced. The list of the top 20 proteins in both controlled CMEs and formulated CMEs are shown in Table 1.

### 3.2. In Vitro Release Study

Release study in pH 7.4 revealed that ARV-825-CME formulation showed significantly (*p* < 0.01) enhanced drug release of 58.93 ± 3.84% at the end of 24 h and 63.25 ± 4.75% at the end of 72 h as compared to ARV-825 solution which showed 14.65 ± 1.49% of ARV-825 release at 72 h which is approximately 4.3-fold increase in release (*p* < 0.001). It was also observed that the ARV-825-CME formulation showed 10.4 times higher drug release at the end of 4 h than the ARV-825 solution (Figure 3A). Release study in pH 1.2 showed 38.61 ± 1.21% and 18.15 ± 2.06% drug release from ARV-825-CME formulation and ARV-825 solution, respectively, at the end of 2 h. It was observed that the ARV-825-CME formulation showed 2.13 times increased drug release as compared to the ARV-825 solution in pH 1.2 buffer (Figure 3C).

### 3.3. In Vitro Permeability of ARV-825-CME

Results showed significantly enhanced cell permeation of ARV-825 in the case of CME formulation as compared to control formulation. The apparent permeability of ARV-825-CME was found to be 22.1 × 10^−6^ ± 0.44 cm/s at 2 h, whereas the apparent permeability of ARV-825 solution was found to be 6.9 × 10^−6^ ± 0.32 cm/s at 2 h. It was observed that the apparent permeability of ARV-825 from ARV-825-CME formulation was increased by 3.2-fold (*p* < 0.0001) as compared to ARV-825 solution (Figure 3B).

### 3.4. In Vitro Cytotoxicity Study

Results showed dose-dependent cytotoxicity when the SF8628 DIPG cells and H1975R cells in 2D cultures were treated with ARV-825, ARV-825-CME, and CMEs. ARV-825-CME was more potent with an IC_50_ value of 275.52 ± 10.97 nM as compared to ARV-825, which had an IC_50_ value of 496.87 ± 16.53 nM after post-treating the SF8628 cells for 48 h. Against H1975R cells, ARV-825-CME was more potent than ARV-825 solution by 1.6-fold. CMEs also demonstrated anticancer potential, with approximately 42.2 ± 2.74% of SF8628 cells and 47.74 ± 1.85% of H1975R cells death when treated with 1 × 10^11^ particles/mL. (Figure 4). Furthermore, the cytotoxicity studies clearly showed a synergistic effect between CMEs and CBD.

It was also observed that the IC_50_ values were reduced in 3D spheroids. The IC_50_ value of ARV-825 using ARV-825-CME formulation was reduced by 2-fold in SF8628 cells and 1.7-fold in H1975R cells. CMEs, like 2D cultures, showed cytotoxicity with 28.85 ± 2.74% and 32.57 ± 1.83 cell death with SF8628 and H1975R cells, respectively (Figure 4). Detailed information on IC_50_ values is provided in Table 2.

### 3.5. In Vivo Internalization of SYTO RNASelect Fluorescent Dye-Stained Exosomes in Tumor

To confirm the internalization of exosomes within the tumor, an initial in vivo study was performed utilizing exosomes that were encapsulated with SYTO RNASelect fluorescent dye that emits green fluorescence with the RNA of the exosomes. The tumor in the mouse was administered PBS as a negative control, and it did not display any fluorescence. However, the mouse that was given SYTO RNASelect exosomes exhibited bright fluorescence in the tumor tissue collected 4 h post-administration, and the tumor from the animal collected 24 h post-administration showed a distinguishable decrease in the fluorescence (Figure 5).

### 3.6. In Vivo Pharmacokinetic Study of ARV-825-CME

Our results revealed that the ARV-825-CME formulation showed a C_max_ of 138.11 ± 6.52 µg/mL ARV-825 in plasma as compared to the control ARV-825 solution, which showed a C_max_ of 54.25 ± 3.30 µg/mL. T_max_ was found to be 6 and 4 h for ARV-825-CME and control ARV-825 solution, respectively. Additionally, the plasma concentration of ARV-825-CME was 2.55 times higher than the control after 6 h. Results also showed significantly enhanced bioavailability of ARV-825-CME formulation with AUC_(0–72h)_ of 2656.56 ± 59.64 µg/mL·h as compared to ARV-825 solution (control) which showed 477.30 ± 23.26 µg/mL·h which is 5.56-fold (*p* < 0.0001) increase in AUC compared to its solution suggesting significant improvement in the oral bioavailability of ARV-825 (Figure 6). After employing compartmental modeling, it was found that ARV-825 followed a one-compartment PK model with rapid absorption and distribution. The elimination phase demonstrated a steady decline in plasma levels, indicating moderate clearance. The area under the curve (AUC) reflected proportional exposure relative to the dose administered. The pharmacokinetic profile of the drug supports appropriate daily dosing intervals to sustain therapeutic levels, ensuring minimal risk of accumulation or adverse effects.

### 3.7. ARV-825-CME Absorption and Distribution Study in GIT

A distribution study was carried out to understand the pathway of absorption of exosomes into the systemic circulation. The SYTO RNASelect containing exosomes were highly concentrated in the intestine, and no fluorescence was observed in the stomach. Among all the intestinal sections, the ileum showed the highest fluorescence intensity, followed by duodenum and jejunum. The intestinal tissue sections showed bright fluorescence at 0.5, 1, and 4 h for duodenum, ileum, and jejunum samples, respectively, after administration of stained exosomes (Figure 7A,B). To confirm, ARV-825-CME was orally administered, and the amount of ARV-825 in each of these sections was determined at different time intervals. At similar time points, the fluorescence intensity curves matched the ARV-825 amount curves in all the intestinal sections. The amount of ARV-825 was highest at 0.5, 1, and 4 h for duodenum, ileum, and jejunum samples, respectively, after administration of ARV-825-CME, similar to the fluorescence study. (Figure 7C,D). In 0.5 h, ARV-825 had reached the duodenum, and by the end of 2 h, most of the drug was present in the ileum. The amount of ARV-825 in each tissue section and plasma at all time points is given in Table 2.

### 3.8. Stability Studies

Stability results showed that ARV-825-CME exposed to pH 1.2 acidic buffer had a protein concentration of 5.9 ± 0.1 mg/mL, particle size of 128.5 ± 2.01 nm, and zeta potential of −18.34 ± 0.07 mV. Furthermore, ARV-825-CME formulation exposed to pH 6.8 phosphate buffer showed a protein concentration of 6.0 ± 0.2 mg/mL, particle size of 131.64 ± 3.84 nm, and zeta potential of −15.25 ± 0.14 mV. Although there was a slight decrease in the formulation’s zeta potential, it was still stable, with values between −10 and −20 mV.

## 4. Discussion

The development of effective anticancer therapies is crucial in combating the global burden of cancer. Despite significant advancements in cancer research, the translation of promising drug candidates into clinically effective therapies remains challenging. One major, never-ending hurdle is the poor pharmacokinetic profile of many anticancer agents, including low water solubility, high molecular weight, and susceptibility to degradation in the harsh acidic and enzymatic conditions of the gastrointestinal tract [12,48,49,50]. Consequently, these drugs face hurdles in achieving adequate dissolution and absorption, resulting in suboptimal bioavailability. Moreover, the phenomenon of first-pass metabolism further diminishes oral bioavailability, as drugs absorbed from the gastrointestinal tract undergo enzymatic degradation in the liver before reaching the systemic circulation [51,52]. Additionally, efflux transporters in the intestinal epithelium actively pump drugs out of the cells, further reducing absorption. These intricate physiological and pharmacological barriers collectively contribute to the observed poor oral bioavailability of anticancer drugs.

By addressing these complexities, researchers aim to improve the efficacy and patient compliance of orally administered anticancer therapies, ultimately advancing the landscape of cancer treatment. In recent years, there have been several attempts to use nano-drug delivery systems, such as liposomes, nanoparticles, and exosomes, which have higher efficacies and lower toxicities [53,54,55,56,57,58,59]. Exosomes are extracellular vesicles that play a crucial role in intercellular communication by transferring bioactive molecules, including proteins, nucleic acids, and lipids, between cells. Due to their natural biocompatibility and ability to traverse biological barriers, exosomes have emerged as promising candidates for drug delivery [19,60,61]. In our study, we isolated and characterized CMEs as potential carriers for our drug. CMEs exhibited favorable characteristics, including stability, biocompatibility, and efficient drug loading capacity, rendering them highly suitable for drug delivery applications.

ARV-825, a novel bromodomain and extra terminal domain (BET) protein degrader, has emerged as a promising therapeutic agent in cancer treatment [34,38]. This small molecule compound exerts its anticancer effects by selectively targeting BET proteins, including BRD2, BRD3, BRD4, and BRDT, which play crucial roles in regulating gene expression and cell proliferation. By inducing the proteasomal degradation of BET proteins, ARV-825 disrupts key oncogenic pathways, such as MYC and NF-κB signaling, leading to growth arrest and apoptosis in cancer cells [38,62,63]. Notably, ARV-825 exhibits potent activity against a wide range of cancer types, including hematological malignancies and solid tumors, making it a versatile candidate for precision oncology [37,38,64,65,66]. Furthermore, ARV-825 has shown efficacy in overcoming drug resistance mechanisms commonly observed in cancer therapy, offering new hope for patients with refractory or relapsed disease. Though studies have demonstrated promising results with ARV-825, it has now been dropped from clinical trials mostly because of the off-target effects and poor bioavailability [67,68,69]. In this study, we explored the potential of camel milk-derived exosomes (CMEs) as a novel drug delivery system to overcome these challenges and improve the delivery of ARV-825, as no delivery system has been used for this molecule.

The study employed a comprehensive approach to investigate the potential of exosome-mediated delivery of ARV-825 for cancer therapy. The successful preparation and optimization of ARV-825-loaded camel milk-derived exosomes (ARV-825-CME) were achieved through a series of steps, including solubilization of ARV-825, emulsification, sonication, and ultracentrifugation. Characterization studies confirmed the entrapment of ARV-825 within CMEs, with TEM imaging validating the integrity of the vesicular membranes, which is in accordance with other TEM studies available in the literature [70]. Milk is a rich source of exosomes, and we were able to isolate 10 mL (1.2 × 10^12^ particles/mL) of exosomes from 500 mL of camel milk. Our results demonstrated a high entrapment efficiency of 42.75 ± 5.05% of ARV-825 within CMEs, with a particle size (between 100 and 150 nm) and zeta potential (−10 to −30 mV) conducive to stability and cellular uptake [71,72]. The exosome membranous vesicles showed similar mean diameters in fluorescent and scatter modes during concurrent size measurements, indicating high particle purity.

Western blot analysis confirmed the presence of exosome marker proteins HSP70, Alix, CD81, and CD63. Previous studies have utilized these proteins to characterize CMEs, and our findings are consistent with their findings [28,30,73,74]. Proteomics analysis showed that more proteins were identified for CMEs compared to the cow and goat milk groups (419 vs. 131–133). The protein cargo of the CMEs was distinctly different from the exosomes from the cow and goat milk groups, while the composition of the exosomes from the cow and goat milk groups was more alike. The proteins that are highly expressed in CMEs are XDH, SLC34A2, MFGE8, and PGLYRP1, out of which, MFGE8 which is milk fat globule epidermal growth factor 8 glycoprotein secreted by activated macrophages and immature dendritic cells and is reported to have anti-inflammatory and antioxidant properties [75,76]. On the other hand, PGLYRP1 is pro-inflammatory alone, but in combination with HSP70 of the CMEs, it becomes cytotoxic to tumor cells [77]. PGLYRP1 was initially thought to be cytotoxic by itself for tumor cells and to function similarly to Tumor Necrosis Factor-α (TNF-α); further research showed that PGLYRP1 alone does not possess cytotoxic activity. Instead, it forms a complex with HSP70, and only these complexes exhibit cytotoxicity toward tumor cells [78]. Interestingly, PGLYRP1 alone actually acts as an antagonist, dampening the cytotoxic effects of the PGLYRP1-HSP70 complexes [79]. However, in the western blotting and proteomics data, we found HSP70 in high amounts, and it is possible that the PGLYRP1-HSP70 complexes might also be responsible for CME anticancer activity. This underscores the complexity of the interactions between various molecules in biological systems and the need for thorough investigation to understand their functions accurately. These proteins are responsible for the anticancer activity of CMEs compared to exosomes from goat and cow milk, which is also supported by other researchers [74,80,81]. This suggests that camel milk exosomes provide additional benefits in terms of anticancer response over cow and goat milk exosomes.

Several studies have reported that sonication would alter the size, protein/miR cargo, lipids, and other contents of exosomes [82,83,84,85]. This is likely due to membrane disruption and exchange during sonication. Like electroporation, the mechanical shear force from the sonicator probe compromises the membrane integrity of the exosomes and allows the drug to diffuse into the exosomes during this membrane deformation. Nevertheless, this membrane deformation process in our studies did not significantly affect the membrane-bound proteins. Haney et al. [83] have reported that sonication and extrusion provide the highest catalase loading into exosomes, as compared with freeze/thaw cycles and passive incubation, which yield the lowest drug loading efficiency. In our dataset, the top 20 proteins are still quite similar in the sonicated and non-sonicated groups. From the un-overlapped proteins, we found most of them are located in cytosol or cytoplasm, which is expected.

One of the key challenges in cancer therapy is achieving sufficient drug permeability and cellular uptake to exert therapeutic effects. The in vitro release study in basic pH buffer demonstrated significantly enhanced drug release kinetics from the ARV-825-CME formulation compared to the control ARV-825 solution (10.4-fold at 4 h and 4.3-fold at 72 h). Similarly, a 2.13 times increased drug release was observed in acidic pH from the ARV-825-CME formulation as compared to ARV-825 solution. This improved release profile could be attributed to the encapsulation of ARV-825 within exosomes, protecting the drug from premature degradation and facilitating sustained release, as seen in other studies. Moreover, our pharmacokinetic (PK) results were aligned with the in vitro release study. In vivo PK results showed a significant increase in systemic drug concentration from the ARV-825-CME formulation compared to the ARV-825 solution at the end of 2 h, suggesting that ARV was absorbed in the stomach under acidic conditions. Similar observations were found in the intestinal basic pH, where the ARV-825-CME formulation exhibited significantly higher ARV systemic absorption compared to the ARV-825 solution. Furthermore, consistent with the in vitro release study, the in vivo PK study results also demonstrated significantly enhanced bioavailability from the ARV-825-CME formulation compared to the ARV-825 solution [86,87].

MDCK cells are favored for in vitro permeability studies due to their ability to form tight junctions and polarize in culture, which emulate epithelial barriers and facilitate reliable, high-throughput screening for drug development. Their ease of culture, consistency, and potential for genetic modification to express human transporters make them invaluable for predicting human drug absorption and metabolism [88,89]. Our in vitro permeability studies revealed a significant increase in apparent permeability (3.2-fold, *p* < 0.0001) of ARV-825 when loaded in CME formulations compared to free drug solutions. This observation is supported by other researchers: Liao et al. reported that milk exosomes have an affinity towards intestinal epithelial cells and can cross the intestinal cells through endocytosis [90]. Kahn et al. also revealed that milk exosomes survive gastric digestion, internalize through intestinal epithelia, and maintain their miRNA abundance in them post absorption [91]. This enhanced permeability can be attributed to the ability of CMEs to overcome biological barriers and facilitate efficient drug transport across cell membranes, as seen in these studies. Studies employing in vitro Caco-2 cell models and in vivo models demonstrated that certain bio-enhancers integrated into liposomes may function by interfering with the structure of the cellular lipid bilayer, which facilitates payload uptake or increases the liposomes’ fusion affinity with cell membranes. Another possible mechanism is the opening of tight junctions, which promotes drug absorption through the paracellular route [92,93]. However, our studies suggest that exosomes possess the natural ability to cross the intestinal barrier via multiple mechanisms, which are still being investigated.

Cytotoxicity assays demonstrated potent anticancer activity of ARV-825-CME formulation against two cancer cell lines, SF8628 DIPG, and H1975R cells, underscoring their efficacy in vitro. Diffuse intrinsic pontine glioma (DIPG, SF8628) is a type of brain tumor predominant in children that forms in the brainstem and has no cure [94]. On the other hand, lung cancer remains one of the most lethal cancers in both men and women [95], and H1975R cells (resistant to Osimertinib with EGFR T790M and L858R mutations) were chosen. 3D spheroid models offer a more physiologically relevant platform for assessing drug efficacy compared to traditional 2D cell culture systems. Consistent with our findings in 2D cell cultures, ARV-825-CME formulation exhibited potent cytotoxicity in 3D spheroid models, indicative of their ability to penetrate and induce cell death within tumor microenvironments. The enhanced anticancer activity can be attributed to cytotoxic effects produced by the exosomes. Numerous studies have previously been published that indicate CMEs have anticancer properties. Badawy et al. have demonstrated synergistic anticancer activity of CMEs loaded with tamoxifen and hesperidin against numerous breast cancer cell lines, including MDA-MB-231 and MCF-7 cell lines as well as in vivo in an MCF-7 xenograft mouse model when administered orally [30]. Furthermore, Ali et al. showed that CMEs exhibited selective anticancer activity against pancreatic cancer cells and was not harmful to normal pancreatic cells. The authors concluded that the mechanism by which CMEs displayed antitumor selectivity was apoptosis, which involves caspases, bax, and bcl-2 [96]. In our proteomics data, we identified two proteins (A0A5N4DFA8/CAND1 and A0A8B8RH14/UBE2) in CMEs, which are related to the ubiquitination pathway that ARV-825 uses for BRD4 degradation. It is possible that ARV-825-loaded CMEs associated with these two proteins is involved in the Cereblon-BRD4 degradation pathway. The synergistic effect observed between exosomes and ARV-825 further emphasizes the potential of exosome-mediated drug delivery in overcoming resistance mechanisms and enhancing therapeutic outcomes in complex tumor models.

Translation of novel drug delivery systems from preclinical studies to clinical applications requires comprehensive pharmacokinetic evaluation in animal models. In our in vivo pharmacokinetic studies, which were performed for the first time for ARV-825 exosomes, we observed significantly higher systemic absorption in plasma and bioavailability of ARV-825 from CME formulations compared to free drug solution. The PK parameters were significantly altered, where we saw a 2.55-fold increase in peak plasma concentration, a 4.5-fold increase in half-life, and a 5.56-fold increase in the AUC of ARV-825. This enhanced pharmacokinetic profile can be attributed to the improved stability and prolonged circulation of ARV-825-CME, resulting in increased drug exposure and therapeutic efficacy, as seen previously in other exosome studies [97,98]. Based on the compartmental modeling of the PK data, ARV-825 from the exosomes is assumed to distribute instantaneously upon absorption into the bloodstream. This distribution phase entails ARV-825 quickly reaching equilibrium in tissues and organs, achieving uniform concentration. Subsequently, the drug undergoes elimination primarily through metabolic processes and excretion, following first-order kinetics, where the rate of elimination is proportional to the drug’s concentration in the body. This simplified approach allows for the estimation of key pharmacokinetic parameters such as the drug’s half-life, clearance rate, and volume of distribution, providing crucial insights into dosing strategies and the potential for therapeutic efficacy and safety profiles.

The SYTO RNASelect fluorescence dye-stained exosomes accumulated in the H1975R tumor in mice observed through fluorescence microscopy clearly shows CME translocation across the intestine and biodistribution to the tumor. In our study, the CMEs were non-targeted and may have reached the tumor via enhanced permeation and retention effect. Distribution studies revealed preferential accumulation of ARV-825-CME in the intestinal sections, particularly in the ileum, suggesting efficient absorption of exosomes in the gastrointestinal tract. In vitro studies have been reported earlier stating that milk exosomes are absorbed through intestinal epithelial cells via endocytosis [99,100]. In our distribution studies, we meticulously tracked the movement of exosomes within the gut. We discovered that CMEs, with their cargo, reached the duodenum within just 0.5 h of administration and progressed further to the ileum, another segment of the small intestine, within 1 h. This rapid transit through the gastrointestinal tract strongly suggests that exosomes are swiftly emptied from the stomach, typically within a timeframe of less than 30 min. Remarkably, this observation aligns with our finding of no detectable presence of ARV-825 in blood samples taken between the 0.5–1 h mark post-administration. However, at 4 h, we observed the highest concentration of ARV-825 in the bloodstream. Notably, beyond this time frame, there was no fluorescence observed in either the duodenum or the ileum, indicating that the CMEs were effectively absorbed through the epithelial lining of the intestines within the initial 1–2 h post-oral administration. This insightful revelation not only sheds light on the dynamics of drug distribution within the body but also underscores the efficiency of CME absorption, corroborating findings from the existing literature [99,100]. Our study, in our opinion, is the first attempt to identify the route of absorption and mechanism in vivo compared to other researchers who showed the mechanism in vitro.

The excellent stability of CMEs in an acidic environment of the stomach was reported earlier [101]. Further, Agrawal et al. reported that there was no discernible difference in the size of exosomes (containing paclitaxel) derived from cow milk in the stomach pH, suggesting that milk exosomes could tolerate severe gastrointestinal conditions [87]. Furthermore, because the miRNA and mRNA in bovine milk were contained in exosomes, Izumi et al. showed that they were resistant to both pH and RNAse degradation [102]. Our stability studies confirmed the robustness of the ARV-825-CME formulation in simulated gastric and intestinal environments, further supporting its potential for oral administration. The possible mechanisms for CME absorption are still being investigated in our laboratory but are assumed to follow similar pathways as reported in the literature, which is through the intestinal epithelium.

The successful development of ARV-825-loaded CME formulations holds significant promise for enhancing targeted anticancer therapy. By improving drug solubility, permeability, and bioavailability, CMEs offer a versatile platform for delivering a wide range of anticancer agents, including ARV-825. Furthermore, the biocompatibility and natural origin of CMEs reduce the risk of immunogenicity and adverse effects associated with traditional drug delivery systems, enhancing their clinical translatability. While our study provides compelling evidence for the efficacy of ARV-825-loaded CME formulation in preclinical models, several challenges and opportunities remain to be addressed. Further optimization of formulation parameters, such as drug loading efficiency and release kinetics, will be essential to maximize therapeutic outcomes. Additionally, long-term safety assessments and scale-up production of CME formulations are necessary steps toward clinical translation. Future studies should also explore combination therapies and personalized approaches to leverage the full potential of ARV-825-CME in diverse cancer types and patient populations. Our study demonstrates the potential of camel milk-derived exosomes as effective carriers for improving the delivery of ARV-825 in anticancer therapy. By enhancing drug solubility, permeability, and bioavailability, ARV-825-CME formulation offers a promising strategy for overcoming existing challenges in cancer treatment. Further preclinical and clinical investigations are warranted to validate the safety and efficacy of this novel drug delivery system and its potential to revolutionize cancer therapy. While ARV-825 demonstrated notable efficacy in reducing tumor size across several preclinical investigations, its translation to clinical trials was thwarted by its insufficient effectiveness in human subjects, primarily attributed to its limited bioavailability at the intended site of action. Presently, a cohort of approximately 20 or more PROTACs are in clinical trials, and most of them, like ARV-825, confront similar challenges, accentuating the critical need for strategies aimed at enhancing their bioavailability to unleash their full therapeutic potential [103]. The employment of CMEs emerges as a promising avenue to surmount this hurdle for the majority of PROTACs undergoing clinical evaluation.

## 5. Conclusions

Cancer remains a formidable global health challenge, with mortality rates highlighting the urgent need for effective treatments. Despite extensive research in anticancer drug development, the success rate of new drugs remains disappointingly low, primarily due to issues such as poor pharmacokinetics and low water solubility. In this study, we explored the potential of utilizing exosomes derived from camel milk as a novel delivery system for improving the bioavailability of the anticancer agent ARV-825, which targets the bromodomain and extra terminal (BET) protein family, specifically BRD4, a critical player in cancer development. The rationale behind utilizing exosomes from camel milk lies in their unique properties, including high biocompatibility, stability, resistance to gastrointestinal degradation, and being an anticancer therapeutic per se. Moreover, milk exosomes offer an abundant and cost-effective source compared to exosomes derived from cell cultures. Our study successfully isolated and purified camel milk-derived exosomes and formulated them with ARV-825 using a well-optimized protocol. Overall, our findings highlight the potential of camel milk exosomes as a promising delivery system for improving the pharmacokinetics and therapeutic outcomes of anticancer drugs like ARV-825. Further preclinical and clinical studies are warranted to validate the efficacy and safety of this novel drug delivery approach in cancer therapy. Moreover, exploring the molecular mechanisms underlying the enhanced anticancer activity of ARV-825-CME formulation and its potential for overcoming drug resistance mechanisms would be valuable for future research efforts in this field.

## Figures and Tables

**Figure 1 pharmaceutics-16-01070-f001:**
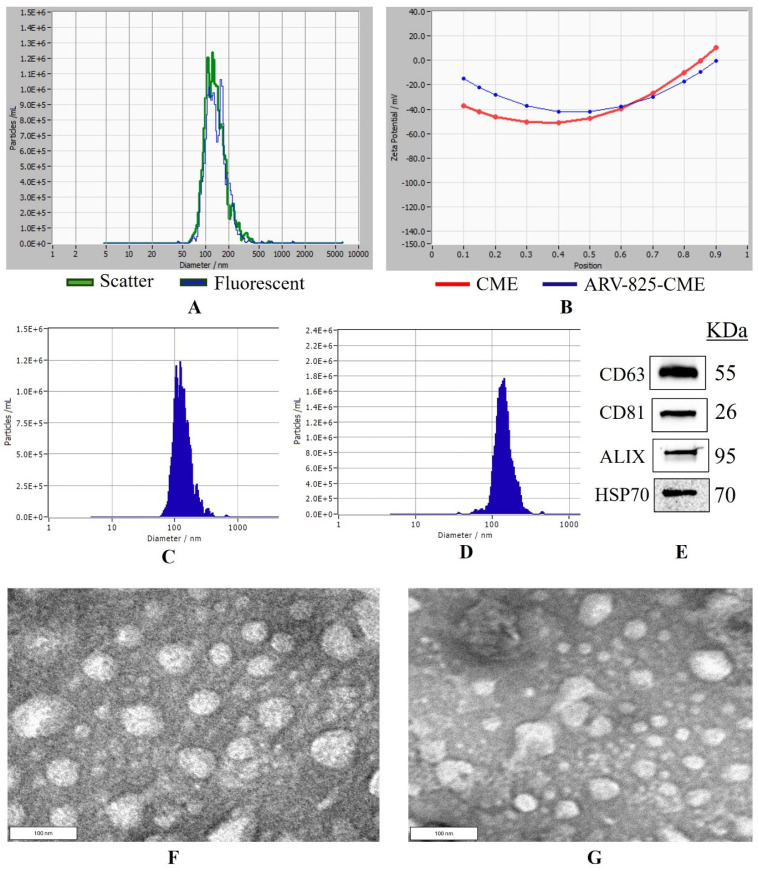
Nanoparticle tracking analysis and Western blotting of exosomes. (**A**) Overlap of particle size distribution graphs of exosomes in scatter and fluorescence mode. (**B**) Zeta potential curve of exosomes and ARV-825-CME. (**C**) Particle size distribution graph of control exosomes before formulation. (**D**) Particle size distribution graph of ARV-825-CME. (**E**) Western blots analysis of exosomes showing expression of proteins HSP70, Alix, CD63 and CD81. (**F**) TEM image of CME, (**G**) TEM image of ARV-825-CME.

**Figure 2 pharmaceutics-16-01070-f002:**
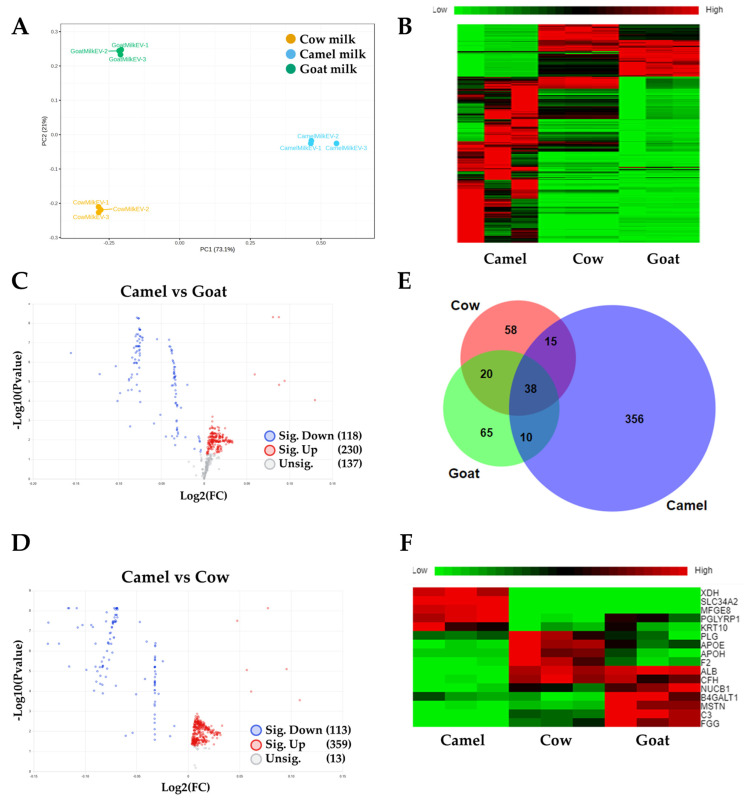
Proteomics analysis of protein cargo of CMEs. (**A**) Principal component analysis (PCA) plot for CMEs compared to Cow and Goat milk; (**B**) Heatmap of differentially expressed proteins (DEPs); (**C**) Volcano plot of DEPs for Camel milk group compared to the Goat milk group; (**D**) Volcano plot of DEPs for Camel milk group compared to the Cow milk group; (**E**) Venn Diagram of all the identified proteins in CMEs compared to Cow and Goat milk. (**F**) Heatmap of overlapped proteins.

**Figure 3 pharmaceutics-16-01070-f003:**
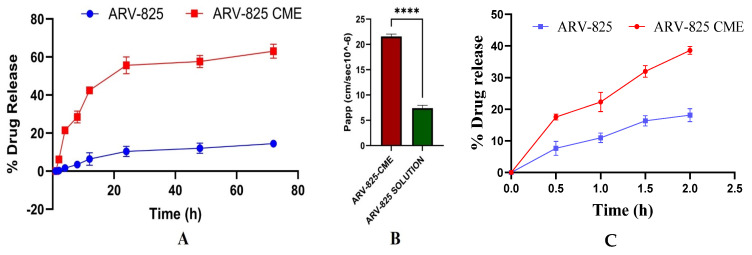
(**A**) An in vitro release study in a pH 7.4 release medium showed significantly increased drug release from the ARV-825-CME formulation compared to the control ARV-825 solution placed in a dialysis bag. (**B**) In vitro cell permeability assay showing enhanced apparent permeability coefficient (Papp) of ARV-825-CME as compared to ARV-825 solution (control). Results were expressed as Mean ± SD (*n* = 3). **** *p* < 0.0001. (**C**) In vitro release study in pH 1.2.

**Figure 4 pharmaceutics-16-01070-f004:**
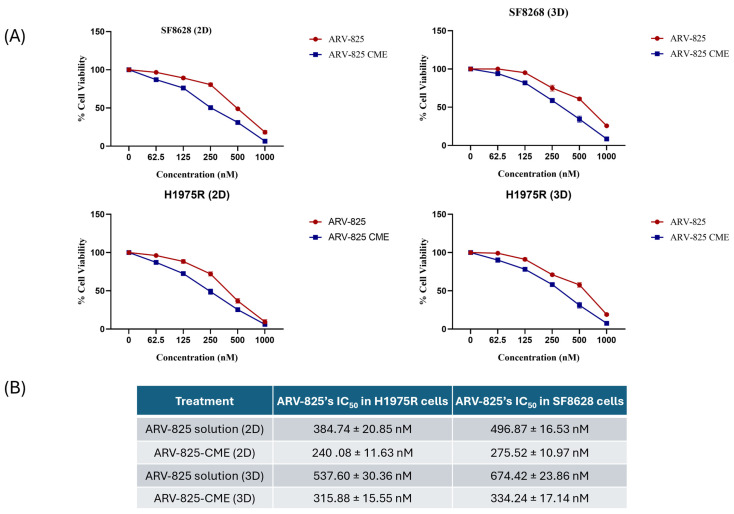
(**A**) In vitro cytotoxicity studies. Line graph showing concentration-dependent cell viability of H1975R lung cancer cells resistant to Osimertinib and SF8628 DIPG cells when treated with CMEs (0–2 × 10^11^ particles/mL) in 2D and 3D cultures. Results were expressed as % in Mean ± SD (*n* = 4). (**B**) ARV-825’s IC_50_ in H1975R and SF8628 cells in 2D and 3D cultures when treated with ARV-825 solution and ARV-825-CME formulation.

**Figure 5 pharmaceutics-16-01070-f005:**
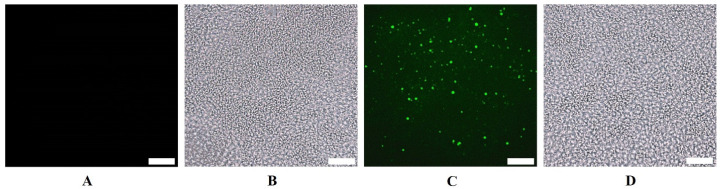
Microscopic (cryosection) images of H1975-R tumors showing internalization of exosomes by the tumor 4 h after oral administration in mice of (**A**) PBS, fluorescent, (**B**) PBS, brightfield, (**C**) SYTO RNASelect stained exosomes, fluorescent, and (**D**) SYTO RNASelect stained exosomes, brightfield. The scale bar represents 10 µm.

**Figure 6 pharmaceutics-16-01070-f006:**
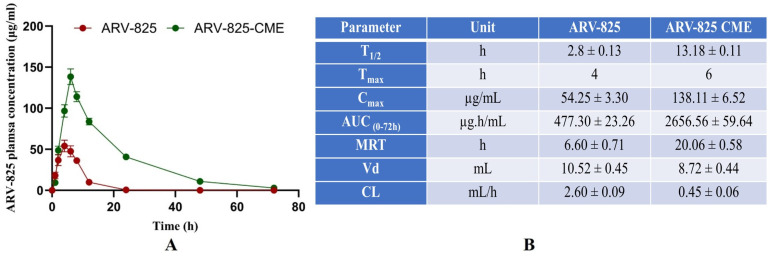
(**A**) The pharmacokinetic study graph shows significantly increased T_1/2_, T_max_, C_max_, AUC, and MRT in the ARV-825-CME group compared to the control ARV-825 solution group. (**B**) Table showing the PK data for parameters T_1/2_, T_max_, C_max_, AUC, MRT, Vd, and CL for ARV-825 solution vs. ARV-825-CME. Results were expressed as Mean ± SD (*n* = 5).

**Figure 7 pharmaceutics-16-01070-f007:**
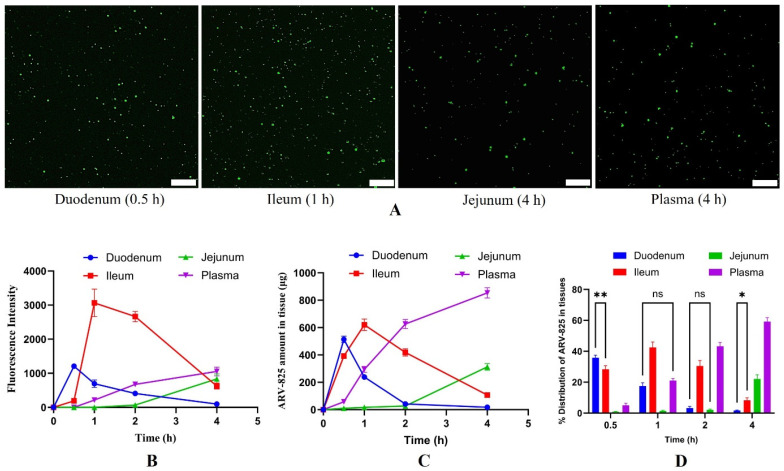
(**A**) Microscopic fluorescent images of intestinal tissue sections showing bright fluorescence at 0.5, 1, 4, and 4 h for duodenum, ileum, jejunum, and plasma samples, respectively, after administration of stained exosomes. The scale bar represents 20 µm. (**B**) Graph showing fluorescence intensities of intestinal and plasma samples from 0 to 4 h after administration of stained exosomes. (**C**,**D**) Graphs showing ARV-825 amount (**C**) and distribution of ARV-825 (**D**) in intestinal and plasma samples from 0 to 4 h after administration of stained ARV-825-CME. Results were expressed as Mean ± SD (*n* = 3). ** *p* < 0.01, * *p* < 0.05, ns is ‘not significant’, *p* < 0.001 (if not mentioned in the bar graph (**D**)).

**Table 1 pharmaceutics-16-01070-t001:** List of top 20 proteins in control CMEs and formulated CMEs after sonication determined by proteomic analysis. Proteins in bold are overlapped.

Control CMEs	Formulated CMEs
**Albumin**	**Lactadherin**
Beta-casein	**Butyrophilin subfamily 1 member A1**
Alpha-S1-casein	**Fatty acid-binding protein, heart**
**Lactadherin** **Butyrophilin subfamily 1 member A1**	**Albumin** **Fatty acid synthase**
**Alpha-lactalbumin**	**Platelet glycoprotein 4**
**Lactotransferrin**	**Xanthine dehydrogenase/oxidase**
Alpha-S2-casein	**Mucin-16**
**Fatty acid-binding protein, heart**	**Sodium-dependent phosphate transport protein 2B**
**Fatty acid synthase**	**Lactotransferrin**
Glycosylation-dependent cell adhesion molecule 1	Ovostatin-like protein 2
**Ig gamma-3 chain C region**	Moesin
**Sodium-dependent phosphate transport protein 2B**	**Alpha-lactalbumin**
**Platelet glycoprotein 4**	Alpha-2-macroglobulin-like protein
**Xanthine dehydrogenase/oxidase**	Ezrin
**Actin**	Folate receptor beta
**Complement decay-accelerating factor**	**Ig gamma-3 chain C region**
Ovostatin homolog 2-like	Alpha-2-macroglobulin
**Mucin-16**	**Actin**
Na exchange regulatory cofactor NHE-RF1	**Complement decay-accelerating factor**

**Table 2 pharmaceutics-16-01070-t002:** Amount of ARV-825 (µg) in intestinal tissues (duodenum, ileum, jejunum) and plasma at 0.5, 1, 2, and 4 h time points.

Time (h)	Duodenum	Ileum	Jejunum	Plasma
0.5	513.96 ± 16.74	393.74 ± 10.24	10.73 ± 2.94	58.12 ± 3.29
1	238.87 ± 12.85	630.52 ± 32.75	17.01 ± 2.97	296.98 ± 15.35
2	42.64 ± 7.62	419.06 ± 18.76	27.49 ± 9.37	626.53 ± 24.30
4	17.36 ± 3.95	107.26 ± 8.56	312.95 ± 17.68	855.25 ± 25.03

## Data Availability

Data are available upon request to the corresponding author.

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
