# Peer review of "Unlocking the Potential of Camel Milk-Derived Exosomes as Novel Delivery Systems: Enhanced Bioavailability of ARV-825 PROTAC for Cancer Therapy"

_pharmaceutics, 2024, doi:10.3390/pharmaceutics16081070_

Round 1

Reviewer 1 Report

Comments and Suggestions for Authors

The authors described camel milk-derived exosomes (CME) as a delivery vehicle for ARV-825. The authors showed that ARV-825 loaded CME gives better ARV-825 releasing efficiency, higher permeability to cells, and better cellular uptake efficiency. Overall , the work is interesting. However, there are some questionable points needed to be addressed.

1. Figure 1F and G are blurry and it's hard to tell CMEs or ARV-825-CMEs.

2. The authors compared the proteomic analysis of exosomes from different sources. However, the authors didn't explain clearly why CME is better than exosomes from other sources. The same problem remains when comparing protein profiles of CME and drug-loaded CME.

3. Figure 4 and Table 2. It's not common to present IC50 curves in where standard deviations are not shown.  While comparing IC50 of ARV-825 solution and ARV-825-CME on cell lines, the authors should show the IC50 curves comparison rather than giving a table of numbers.

4. Figure 5. Fig 5B and 5D don't look like normal H&E staining images.  Also, while the authors stated in the main text there were fluorescence shifts 4h and 24 h after SYTO RNASelect exosome injection, this is not shown in Figure 5.

Reviewer 2 Report

Comments and Suggestions for Authors

pharmaceutics-3130315

Unlocking the Potential of Camel Milk-Derived Exosomes as 2 Novel Delivery Systems: Enhanced Bioavailability of ARV-825 3 PROTAC for Cancer Therapy.

This is a well written and extensive manuscript which investigates the use of camel milk-derived exosomes as a carrier for anti-cancer agent ARV-825. Camel milk-derived exosomes (CME) were used as they had been reported to have anti-cancer activity themselves increasing apoptosis. The authors characterised the isolated the CME, characterised the CME, loaded the CME with ARV-825 and carried out in-vitro and in-vivo trials on the increased absorption and targeting.

The manuscript is well written.

Minor suggestions and queries

L 416. The different modes are at the end of the sentences with is a little confusing at first. Perhaps start with mode. E.g. In scatter mode, the CME NTA results showed ……  In fluorescent mode, the particle number was …..

L 459 How can about 365 proteins be identified in CME and CME ARV but only 202 overlaps? Presumably CME ARV should be all of CME plus the additional loading on ARV. Does this mean that on loading of ARV some CME proteins are lost and are these lost in proportion or are only specific proteins lost. Presumably CME ARV involves further washing steps. Is this part of the difference, that is non integral proteins of CME are washed away on further processing?

Figure 2. Would be clearer if Figure 2 B and E had legends on the left-hand side. Perhaps DEPs and Overlapped proteins.

Figure 4 Is the X axis of 0*1011 correct? This is zero. Perhaps it is 1 * 1011 and the others need to be adjusted.

Reviewer 3 Report

Comments and Suggestions for Authors

Nathani et al. investigated the use of milk-derived exosomes as a carrier for the oral administration of ARV-825 protac. The study investigated an emerging field of drug delivery that may be of interest to the journal readers.

I have the following observations:

1. The introduction section needs to be improved. It is long, and the significant parts (oral administration, exosome, ARV-825 protein) are not adequately linked together.

2. Justify why MDCK cells have been used for permeability studies. They are renal cells, but the study's exosomes are intended for oral administration. Also, justify why lung cells have been used for cytotoxicity studies.

3. Materials and methods are divided into many paragraphs, which are quite confusing. They should be better organized and supported by a narrative sentence at the end of the introduction section so that what has been performed is clear.

4. It is not clear why the proteomic data indicate that camel milk is the preferred source to isolate exosomes.

5. It is not clear why the control CME and formulated CME after sonication have different proteomic compositions.

6. Release studies should also be performed at pH=1 to simulate what happens in the stomach. All the active may be released before reaching the intestine.

7. Figure 5 is not appropriate to support what is described in the text.

8. Discussions need to be improved. As for the introduction, it is too long and must be more logically organized.

Round 2

Reviewer 1 Report

Comments and Suggestions for Authors

The authors addressed most of my comments and the manuscript is significantly improved.

Reviewer 3 Report

Comments and Suggestions for Authors

I am satisfied with the revisions provided.